# Efficient information extraction using LLMs and knowledge distillation: A study on HPV health communication

Saadat Hasan Khan[1]*, Kevin Lybarger[2]

**1** Department of Computer Science, George Mason University, Fairfax, Virginia, United States of America,
**2** Department of Information Sciences and Technology, George Mason University, Fairfax, Virginia, United States of America

* skhan225@gmu.edu

## Abstract

State Department of Health (DOH) websites serve as authoritative sources of HPV-related health communications, presenting state-specific content that influences public awareness and vaccination decisions. We develop a computationally efficient framework to systematically evaluate these information repositories based on their content quality, completeness, and their motivational impact on vaccination behavior. We propose a dataset consolidating 48 different DOH websites' data targeted towards HPV and HPV vaccination. By developing an annotated dataset (n = 400), efficient prompting techniques and a Knowledge Distillation framework, we develop and evaluate efficient student models based on the Llama family of Large Language Models (LLMs) and the RoBERTa Large encoder architecture. We finally deploy the best-performing student model for a computationally feasible evaluation of the content of DOH websites. We show that fine-tuned RoBERTa Large model achieves an F1 score of 0.74 on the test set, outperforming all other student models and approaching the teacher model's performance (F1 = 0.77). The fine-tuned RoBERTa-Large model is subsequently applied to data from various state DOH websites to evaluate the information presented. We also discuss the broader implications, limitations, and ethical and legal considerations of the proposed approach.

## Author summary

We studied how state health department websites communicate information about human papillomavirus (HPV) and vaccination. These websites are important because they shape public understanding and influence people's decisions about getting vaccinated. We collected information from 48 state websites and looked at how complete, clear, and persuasive their content was. To do this, we created a smaller, efficient computer model that could quickly read and evaluate large amounts of website text. We trained this model using a teaching approach

**Data availability statement:** Our data is made publicly available at: https://github.com/SaadatKhan/Efficient-Information-Extraction-Data and archived via Zenodo: https://doi.org/10.5281/zenodo.18691664.

**Funding:** The author(s) received no specific funding for this work.

**Competing interests:** The authors have declared that no competing interests exist.

where a larger, more powerful model showed it how to make good decisions. Our smaller model was almost as accurate as the larger one, but much faster and easier to use. We then used it to review each state's website. This approach can help public health organizations identify gaps in the information they provide and improve how they communicate important health messages. While we focused on HPV, the same method could be used to study how other health topics are presented online.

## Introduction

Human papillomavirus (HPV) is a leading cause of cancer in the United States, responsible for approximately 48,000 cases annually, with cervical cancer being most prominent [1]. Although the HPV vaccine demonstrates high efficacy (88%) in preventing HPV-induced cervical cancers, vaccination rates in the U.S. remain suboptimal, with only 76.8% of adolescents and 39.9% of adults aged 18–26 having received the vaccine [2–4]. In contrast, Australia achieved 82% HPV vaccine uptake among 12-year-old girls and is projected to eliminate cervical cancer within 20 years [5]. Multiple factors contribute to lower U.S. HPV vaccination rates, including state policies, socioeconomic factors, cultural attitudes, and most importantly, public awareness levels [6].

Public awareness and HPV vaccine uptake are influenced by a complex information ecosystem including social media, news, public health communications, and state-sponsored Department of Health (DOH) websites [7,8]. Prior studies have shown that effective communication strategies have proven to be advantageous in vaccination uptake rates for various diseases [9,10]. As authoritative sources for communicating information, DOH websites provide verified information but vary significantly in quality, comprehensiveness, and accessibility [11,12]. Automated, scalable analysis of online health information is critical for assessing content quality and completeness. For HPV vaccination, state DOH websites serve as key sources yet vary widely in content. Evaluating these sites can help identify dissemination gaps, improve awareness, and reduce health disparities.

Online health resources, including DOH websites, present information primarily as unstructured text [13,14]. While manual content analysis is labor-intensive and lacks scalability, Natural Language Processing (NLP) techniques enable automated, large-scale analysis through text classification, and Information Extraction (IE) methodologies like Named Entity Recognition (NER), and Relation Extraction (RE) [15–19]. Although these methods have been applied to HPV topics in social and news media to analyze vaccination stance, sentiment, and public opinions, no comprehensive analysis has systematically identified, extracted and categorized HPV information patterns and knowledge structures across DOH websites [20–25].

In general, characterizing online health information at scale is challenging due to the need to derive structure from unstructured text and resolve semantic ambiguities

[26–30]. These challenges apply directly to assessing the quality of HPV-related content on DOH websites, where information is largely unstructured and semantically variable. NLP techniques have evolved from rule-based systems with poor generalization to statistical methods lacking deeper language understanding [31–34]. Neural architectures like Recurrent Neural Networks (RNNs) advanced semantic modeling but required extensive training data and struggled with long contexts [35–39]. Instruction-tuned LLMs, through in-context learning and chain-of-thought (CoT) prompting, perform well with limited data, reducing task-specific training data needs and making them a suitable solution for evaluating information quality in DOH websites [39–44].

Knowledge Distillation (KD) addresses computational barriers to deploying LLMs for IE tasks in online health information [45]. While models like DistilBERT and TinyBERT have demonstrated KD's effectiveness for model compression, recent advances extend KD to distilling reasoning capabilities from teacher LLMs to smaller student models [46–48]. Among these advances are CoT distillation techniques, which focus on transferring complex LLM reasoning capabilities to smaller models [49,50]. Applying KD to health IE tasks remains challenging due to model compatibility issues, domain-specific knowledge transfer, and performance robustness, but it represents a promising direction to balance computational efficiency and annotation needs [51–53].

This study proposes a generalizable KD framework for health IE, designed to enable scalable, high-performance inference using compact models with minimal annotation requirements. While the approach is broadly applicable to diverse health communication settings, we evaluate its effectiveness in the context of HPV-related content on state DOH websites, a domain characterized by unstructured text and significant variability in information quality. Specifically, we investigate how the performance of a compact encoder-only model (RoBERTa Large) compares to LLMs in classifying key health topics Our framework integrates task-specific annotation, prompting strategies, and KD-based model compression to transform unstructured content into structured, semantically coherent representations suitable for downstream evaluation. Beyond HPV, this approach can be readily applied to assess the quality, completeness, and clarity of online health information in other domains such as influenza, COVID-19, chronic disease prevention, and emerging public health threats.

In summary, the key contributions in our research are:

- Generalizable KD framework for distilling reasoning capabilities from LLMs to compact encoder-only models for health IE tasks

- Development of a structured Q&A schema transforming unstructured health content into semantic representations for HPV information evaluation

- Creation of a benchmark dataset with 400 doubly-annotated samples for model evaluation

- Case study analyzing over 14k paragraphs across 48 DOH websites evaluating topic coverage and quality

## Results

We present experimental results in four parts, reflecting the major stages of our study. First, we compare prompting strategies across multiple large LLMs to identify the most effective teacher model for KD. Next, we evaluate the performance of smaller student models distilled from this teacher under different KD approaches. Each of these results are statistically validated by performing pairwise bootstrap significance testing. We then conduct an error analysis to examine differences in performance between teacher and student models, highlighting strengths and weaknesses in explicit versus inferred information extraction. Finally, we apply the best-performing student model to the Comprehensive HPV Corpus to assess information coverage across 48 DOH websites. More details about our experimentation methods are presented in the "Materials and Methods" section of this paper.

## Prompting strategies

We evaluated multiple models using the proposed prompting strategies. Initial tests with Zero-shot + Guidelines were conducted with both Llama models and GPT-4o. GPT-4o underperformed compared to Llama 3.1 70B (0.55 F1), making it unsuitable as a teacher model for KD. Additionally, OpenAI's terms of service prohibit using their models to train other models, so GPT-4o was excluded from further experimentation [54].

Table 1 presents performance across the Llama family using prompt-based inference without supervised fine-tuning. Zeroshot + Guidelines with Llama 3.1 70B achieved 0.77 F1. While 1-shot + CoT improved smaller models' performance (Llama 3.1 8B and Llama 3.2 3B), it decreased Llama 3.1 70B's performance. Across all prompting strategies, Llama 3.1 70B consistently outperformed other models. Table 1a demonstrates that Llama 3.1 70B with Zero-shot + Guidelines prompting (marked with * in Table 1a) significantly outperforms all other candidate teacher models.

## Knowledge distillation

Table 1 summarizes student model performance across KD approaches. For each experiment, Llama 3.1 70B was the teacher model under different prompting strategies. The results show the KD pipelines consistently improved smaller

**Table 1. Performance comparison of models on prompt-based experiments and knowledge distillation. Significance: * p < 0.05 (Prompt-based), ♠ p < 0.05 (KD framework).**

**a) Performance of LLMs in prompt-based experiments**

| Experiment Type | Model | P | R | F1 |
|---|---|---|---|---|
| Zero-shot + Guidelines | Llama 3.2 1B | 0.08 | 0.73 | 0.14 |
| | Llama 3.2 3B | 0.37 | 0.30 | 0.33 |
| | Llama 3.1 8B | 0.55 | 0.52 | 0.54 |
| | Llama 3.1 70B | **0.75** | **0.79** | **0.77**\* |
| Zero-shot + Guidelines + Rationale | Llama 3.2 1B | 0.13 | 0.08 | 0.10 |
| | Llama 3.2 3B | 0.51 | 0.35 | 0.41 |
| | Llama 3.1 8B | 0.60 | 0.43 | 0.50 |
| | Llama 3.1 70B | 0.66 | 0.78 | 0.71 |
| 1-shot + CoT | Llama 3.2 1B | 0.07 | 0.49 | 0.12 |
| | Llama 3.2 3B | 0.38 | 0.58 | 0.46 |
| | Llama 3.1 8B | 0.55 | 0.84 | 0.66 |
| | Llama 3.1 70B | 0.68 | 0.77 | 0.72 |

**b) Performance of student models in KD-based experiments**

| Distillation Type | Model | P | R | F1 |
|---|---|---|---|---|
| Encoder Baseline | RoBERTa | 0.72 | 0.77 | **0.74**♠ |
| | DistilBERT | 0.68 | 0.54 | 0.61 |
| Encoder + CoT | RoBERTa | 0.70 | 0.74 | **0.72**♠ |
| | DistilBERT | 0.70 | 0.53 | 0.60 |
| LLM Baseline | Llama 3.2 1B | 0.08 | **0.76** | 0.14 |
| | Llama 3.2 3B | 0.36 | 0.31 | 0.33 |
| | Llama 3.1 8B | 0.52 | 0.61 | 0.56 |
| LLM + Rationale | Llama 3.2 1B | 0.21 | 0.06 | 0.10 |
| | Llama 3.2 3B | 0.50 | 0.34 | 0.41 |
| | Llama 3.1 8B | **0.79** | 0.37 | 0.50 |
| LLM + CoT | Llama 3.2 1B | 0.07 | 0.48 | 0.12 |
| | Llama 3.2 3B | 0.25 | 0.62 | 0.35 |
| | Llama 3.2 8B | 0.58 | 0.75 | 0.65 |

student models, except Llama 3.2 1B. The Encoder Baseline with RoBERTa achieved best performance at 0.74 F1, only 0.03 F1 below the teacher model (0.77 F1). Furthermore, significance testing revealed no statistically significant difference between the student and teacher models, suggesting that the student's strong performance was unlikely to have occurred by chance. Incorporating CoT labels to RoBERTa (Encoder + CoT) provided no improvement over Encoder Baseline. Similarly, for LLM students, neither rationales (LLM + Rationale) nor supervised CoT (LLM + CoT) offered performance gains.

Table 2 presents a detailed comparison between the teacher model (Llama 3.1 70B with Zero-shot + Guidelines) and best-performing student model (RoBERTa with Encoder Baseline). The student model outperforms the teacher model on five labels and achieves near-parity (gaps under 0.06 F1) in four additional labels. These results demonstrate effective knowledge transfer in KD. The strong performance of the student, coupled with significantly reduced computational requirements, highlights the utility of KD in developing resource-efficient models for large-scale information assessment. Our experiments reveal that the inclusion of additional CoT information (Encoder + CoT) provided no performance improvement over the simpler Encoder Baseline. Given the equivalent or superior performance, simpler architecture, we select RoBERTa with the Encoder Baseline as our final student model. Henceforth, "student model" refers exclusively to RoBERTa with Encoder Baseline, and "teacher model" to Llama 3.1 70B with Zero-shot + Guidelines.

## Error analysis

The teacher model outperformed the student model in identifying labels associated with straightforward, explicitly stated information over those needing nuanced understanding. The teacher model achieved perfect F1 scores (1.00) for No Cure and Number Doses labels, and strong performance (0.80 F1) for Highly Effective. The teacher model excels at extracting explicit information (e.g., "HPV has no cure...", "2 vaccination doses are required...") rather than processing subtle or indirect content, demonstrating stronger capability in straightforward information extraction compared to nuanced interpretation. This finding parallels the 'overthinking' problem many advanced LLMs suffer from where excessive reasoning can harm performance [55]. In contrast, the student model performed better on labels requiring inference from less explicit language. It outperformed the teacher model on labels like Cancer Prevention (0.85 vs 0.73 F1), Sexual Spread (0.92

**Table 2. Performance comparison between teacher (Llama 3.1 70B) and student (RoBERTa) models across different labels.**

| Label | Number of Instances | Llama 3.1 70B (Teacher) | | | | RoBERTa(Student) | | | |
|---|---|---|---|---|---|---|---|---|---|
| | | P | R | F1 | σ² | P | R | F1 | σ² |
| Asymptomatic | 18 | 0.71 | 0.94 | 0.81 | 0.09 | 0.60 | 0.83 | 0.70 | 0.08 |
| No Cure | 7 | 1.00 | 1.00 | 1.00 | 0.02 | 0.75 | 0.43 | 0.55 | 0.01 |
| Sexual Spread | 28 | 0.81 | 0.89 | 0.85 | 0.10 | 0.87 | 0.96 | **0.92** | 0.11 |
| Cause of Cervical Cancer | 19 | 0.85 | 0.58 | 0.69 | 0.05 | 1.00 | 0.63 | **0.77** | 0.05 |
| Cancer Prevention | 21 | 0.64 | 0.86 | 0.73 | 0.10 | 0.77 | 0.95 | **0.85** | 0.08 |
| Cause of Cancer | 51 | 0.80 | 0.86 | 0.83 | 0.15 | 0.76 | 0.88 | 0.82 | 0.17 |
| Prevent Spread | 20 | 0.58 | 0.55 | 0.56 | 0.09 | 0.57 | 0.85 | **0.68** | 0.11 |
| Highly Effective | 15 | 0.80 | 0.80 | 0.80 | 0.05 | 0.64 | 0.47 | 0.54 | 0.04 |
| Number Doses | 9 | 1.00 | 1.00 | 1.00 | 0.03 | 1.00 | 0.89 | 0.94 | 0.04 |
| Side Effects | 9 | 0.71 | 0.56 | 0.63 | 0.02 | 0.71 | 0.56 | 0.63 | 0.02 |
| Rec-Children | 21 | 0.68 | 1.00 | 0.81 | 0.11 | 0.57 | 0.95 | 0.71 | 0.11 |
| Rec-Male Female | 6 | 0.83 | 0.83 | 0.83 | 0.02 | 0.33 | 0.17 | 0.22 | 0.01 |
| Stance-Recommend | 24 | 0.79 | 0.79 | 0.79 | 0.07 | 0.78 | 0.58 | 0.67 | 0.07 |
| Stance-Optional | 9 | 1.00 | 0.22 | 0.36 | 0.01 | 1.00 | 0.33 | **0.50** | 0.01 |
| Source Credibility | 25 | 0.69 | 0.72 | 0.71 | 0.10 | 0.66 | 0.76 | 0.70 | 0.01 |
| **Micro Average** | | 0.75 | 0.79 | 0.77 | | 0.72 | 0.77 | 0.74 | |

vs 0.85 F1), and Prevent Spread (0.68 vs 0.56 F1). These labels often contain information embedded within broader narratives or presented indirectly, requiring inference rather than explicit fact extraction. Examples include "These are all viruses that can be passed during sex..." and "Diseases and Infections related to HPV declined since the introduction of HPV vaccine..." This pattern suggests that model compression through distillation may preserve or even enhance capabilities for **interpreting implicit information** while reducing performance on explicit fact extraction. To provide further clarity, complete examples of the data representing such classes are included in the S3 Text.

These patterns also extend to recommendation-related labels. The teacher model excels with explicitly presented information (e.g., "Doctor recommends..." or "It is recommended for children..."). In contrast, for Stance-Optional (determining if the vaccine is presented as optional), the student model outperforms the teacher (0.50 vs 0.36 F1). However, some notable discrepancies exist in certain cases, such as the student model significantly underperforming on Rec-Male Female (0.22 vs 0.83 F1), suggesting insufficient transfer of sex-related medical reasoning during KD. These inconsistent performances across similar labels indicate that success depends less on the question type and more on how information is structured in text samples.

### Case study with the comprehensive HPV Corpus

We applied the student model to the entire Comprehensive HPV Corpus, analyzing all paragraph-level entries for 15 predefined labels. Model outputs were converted to one-hot encodings for each paragraph, then aggregated by DOH website to count unique labels appearing at least once per site. This representation captures topic coverage rather than frequency, addressing the observation that states often repeat information across multiple pages. This approach provides a meaningful assessment of how comprehensively each state covers HPV topics.

Fig 1 displays a heat map of HPV information density across states, with darker colors indicating higher topic coverage and lighter colors showing lower coverage. Of the 50 states, 48 had web-scrapable content, while two (shown in gray) were excluded due to data restrictions or absence of information. The numeric overlay on each state represents how many of the 15 predefined labels were detected on that state's DOH website. Despite Google Search API limitations, some states demonstrated comprehensive coverage because their HPV information was concentrated on a few key pages. These high relevance pages were successfully identified by the API's algorithm's preference on relevance, validating the methodological choice even with constraints on the number of returnable results. Additional experiment for validating this claim of covering adequate content despite the 100 page limit from Google Search API is presented in S1 Text. The figure also shows that no state covers all 15 points of interest, yet no single label is universally missing, validating the label construction. Five states (New Mexico, New York, North Dakota, Texas and Washington) include 14 labels, 31 states have at least 10, and only 5 states have fewer than 5 target information points, highlighting significant variation in HPV information coverage across jurisdictions. For states with low coverage (information points), this was not attributable to API limitations, as the number of pages returned for these states was well below 100.

### Discussion

This study examines various KD strategies using LLMs and encoder-only models to evaluate health information online, focusing on HPV content on DOH websites. It addresses the need for efficient public health information analysis. Results show that with effective KD, a small RoBERTa model can analyze HPV content with performance comparable to Llama 3.1 70B. In this section, we delve further by discussing the system performance, practical applications, broader implications, limitations, and ethical considerations of such a system.

### Performance interpretation

The KD experiments reveal key insights about model architecture and knowledge transfer. RoBERTa achieved the highest student model performance (0.74 F1), nearly matching the teacher model (0.77 F1). This success stems from RoBERTa's encoder-only architecture being well-suited for classification tasks, enabling effective learning from teacher predictions.

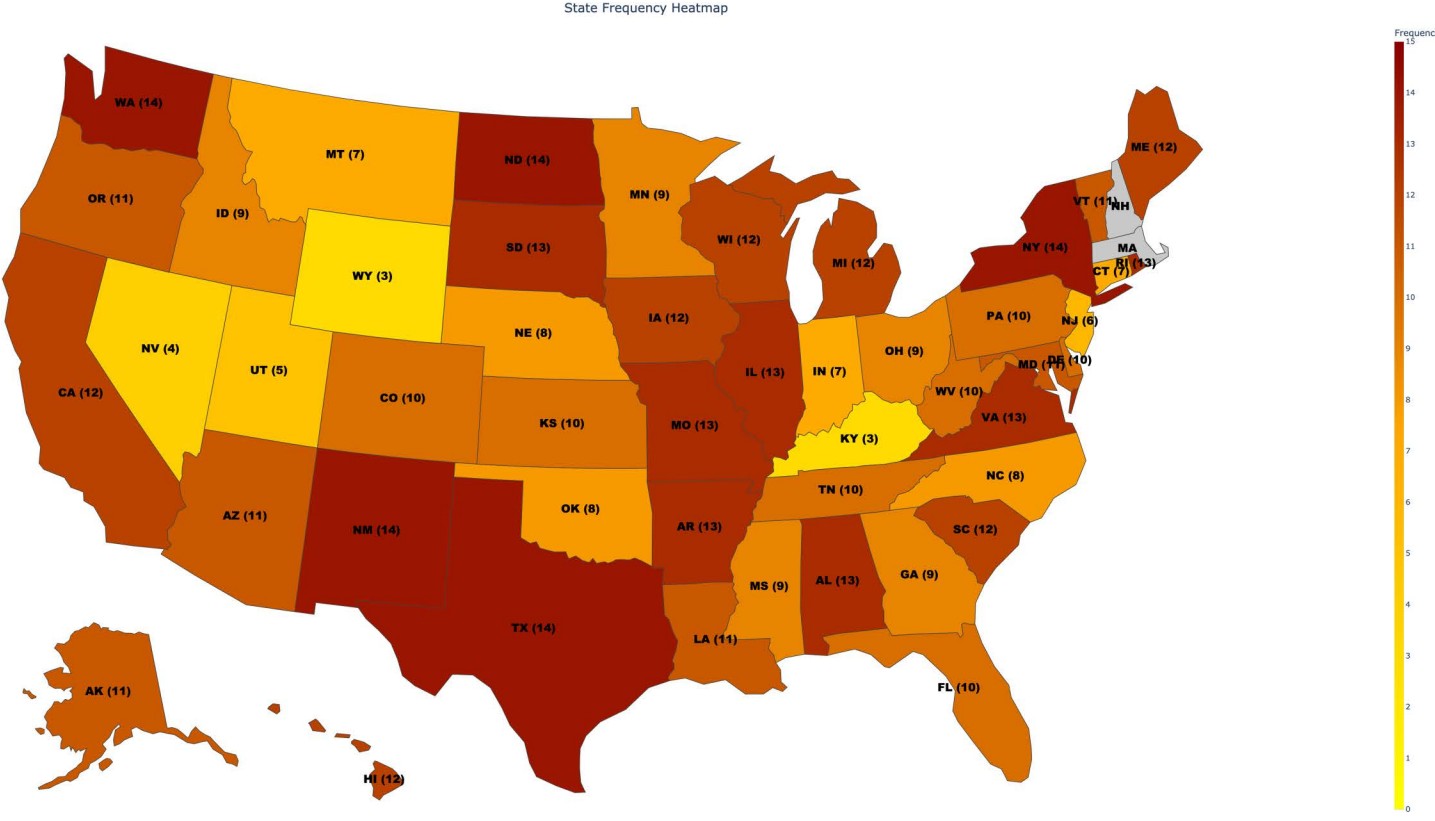

**Fig 1. Heatmap representing density of information of each States' DOH websites.** Base map generated with Plotly python package with geometry sourced from Natural Earth.

While LLMs excel in generative tasks, the experiments show that distilling teacher knowledge for classification through a QA framework is more challenging for them. The multi-label classification approach aligns better with RoBERTa's architectural strengths than with autoregressive capabilities of generative models, explaining RoBERTa's superior performance despite its smaller parameter count. Furthermore, our empirical results show simpler distillation approaches (Encoder Baseline) outperform more sophisticated frameworks with CoT questions or rationales for most models. This suggests streamlined knowledge transfer is more effective for this task, as additional reasoning steps may introduce unnecessary complexity and impair generalization from training data.

A runtime analysis comparing student and teacher models used identical resources (1 NVIDIA A100 GPU) on the validation set. The teacher model processed samples serially through the QA format, while the student model handled all 15 labels simultaneously through multi-class classification. Results showed that the student model, with a 200-fold reduction in parameters, performed inference approximately 350 times faster. This efficiency, combined with comparable performance, indicates lightweight student models offer scalable solutions for large-scale, continuous health information monitoring.

## Broader implications

The KD pipeline extends beyond post-hoc evaluation to enable real-time monitoring of health information across digital ecosystems, allowing for continuous assessment and timely alerts to support rapid public health communication responses. The system is not limited to communicable diseases but is equally applicable to non-communicable diseases

(NCDs), such as diabetes or chronic pain, where the dissemination of accurate and high-quality information is essential for effective disease management and patient decision-making. The system can be used to assess the quality of health information disseminated across platforms and to examine the correlation between the presence of authoritative data and practical outcomes, such as vaccination uptake rates. The framework is also adaptable to various health topics and sources including news and social media as it can be leveraged to identify information gaps and inconsistencies through trained pattern recognition models, providing guidance for repository improvements. Lastly, its scalable framework makes it suitable for quality assessment across diverse health topics and ecosystems.

## Limitations of the proposed work

This work has some limitations which inform future research. Primarily, data sources is one limitation. While DOH websites provide authoritative information, they represent only a portion of the health information ecosystem and lack the misinformation complexity found in less regulated channels like news and social media. Analyzing DOH websites alone does not fully capture state-specific health policies or local health department communications distributed through other mediums, limiting our understanding of state government health information dissemination.

Furthermore, the data collection was limited by the fragmented, state-independent dissemination of HPV-related information, complicating both manual and Google Search API approaches and risking content oversight. Additionally, focusing solely on website text excluded information in PDFs, videos, and interactive media, potentially omitting important elements needed for comprehensive evaluation.

Lastly, the analysis of English-only content limits findings given U.S. linguistic diversity. Though DOH websites offer multilingual versions, we did not assess content parity or potential information asymmetries. Additional work is needed to understand the information landscape for non-English-speakers.

## Ethical and legal considerations

Pre-trained language models may reflect biases present in large-scale training data, including social, cultural, and demographic prejudices [56,57]. In the health domain, such biases could influence how specific topics are represented or interpreted [58,59].While our work focuses on content-level evaluation rather than operational deployment, we acknowledge so Multiple potential sources of bias that could propagate if the KD pipeline is integrated into downstream systems.

First, variations in reading level, linguistic complexity, and cultural framing of health content may affect model predictions in ways that disadvantage certain populations [60]. Second, disparities in digital access vary across populations, and any deployed system should be designed with these inequities explicitly in mind [61]. Third, information quality varies across webpage hierarchies (primary vs. secondary pages), introducing potential sampling bias. We attempted to sample webpages uniformly, though future work should develop systematic methods to account for these quality gradients.

We provide initial quantitative assessment of bias risk through analysis of differential model performance across label categories. Our results show that explicit fact-based labels achieve stronger performance than inference-based labels, indicating that certain health information types are classified more reliably than others. While we do not systematically assess the linguistic and access-related factors noted above, these sources of variation could meaningfully affect evaluations of health information quality. If deployed operationally, bias propagation from all these sources should be systematically examined and mitigated.

The data collection was restricted to publicly available information from DOH websites. However, some states impose restrictions on automated data collection, which constrained our ability to collect data uniformly from all jurisdictions. The annotated Targeted HPV Dataset consists of 400 paragraphs and is intended solely for academic research. It is available to promote reproducibility, without affecting the market value or integrity of the original content.

## Materials and methods

### Data

We collected two datasets: (1) Targeted HPV Dataset, consisting of 400 annotated paragraphs from DOH websites related to HPV for model evaluation, and (2) Comprehensive HPV Corpus, containing over 14k unlabeled paragraphs retrieved via the Google Search API. Details on sampling and preprocessing are provided below. Throughout, we refer to individual paragraphs as "samples." All data used in the subsequent experiments were collected prior to December 2024.

### Targeted HPV dataset

We constructed an annotated dataset by targeting DOH webpages that frequently contain HPV-related information of interest, using a two-tier hierarchical crawling strategy. First, we manually identified primary HPV-specific pages for each state's DOH domain. We then automatically retrieved secondary pages by following outbound links from primary pages, including only those containing the terms "HPV" or "Human papillomavirus". This process yielded 41 primary and 1,124 secondary pages. Some states lacked HPV-specific pages or imposed security restrictions that prevented data collection.

Preliminary analysis showed that information relevance declined sharply when extending link traversal beyond one degree from the primary pages. While additional levels increased the number of pages, they rarely contained HPV content. Although secondary pages contained more text on average (227.46 paragraphs vs. 20.56 on primary pages), much of it consisted of fragmented or low-relevance content, such as navigation menus or brief, unrelated sections, rather than substantive HPV-related information. To optimize content relevance, data collection was limited to primary pages and directly linked secondary pages, resulting in a higher concentration of meaningful content compared to broader but noisier collections produced by automatic search APIs. These samples were then annotated to establish a gold standard reference. The inclusion of secondary webpages helps in creating a more balanced dataset for better evaluation.

We randomly sampled 400 paragraphs (313 from primary pages, 87 from secondary) for annotation, maintaining higher proportion from primary pages to ensure sufficient concentration of HPV content. These annotated samples comprise the Targeted HPV Dataset, which was split into a development set (33%, 132 samples) and test set (67%, 268 samples) for evaluating IE strategies.

### Comprehensive HPV corpus

While the Targeted HPV Dataset captured key phenomena with a balanced representation of positive and negative cases, its manual page selection risked omitting relevant content. To ensure broad coverage, we created the Comprehensive HPV Corpus using the Google Search API, querying each state with terms "HPV" and "Human papillomavirus" and retrieving top 100 results per state. The data were collected exclusively from publicly available and web-scrapable only sources. This approach was appropriate as HPV content typically includes these terms explicitly. We excluded attachments (e.g. PDFs, PowerPoints, and spreadsheets) and external links, retaining only text from state DOH domains. The final corpus comprises 14,444 samples from 48 states.

Data collection was constrained by state-specific access restrictions: Massachusetts and New Hampshire returned HTTP 403 (Forbidden) errors due to security measures preventing programmatic extraction and were therefore excluded from the case study.

The 100-result limit of the Google Search API posed challenges for states where HPV information spanned more than 100 pages. This constraint was exacerbated where HPV terms appeared mainly in navigation elements rather than informative content, resulting in sparser coverage. While the 100-result limit could theoretically constrain coverage, Google's relevance-based ranking prioritizes the most substantive pages [62]. S3 Text shows analysis of North Dakota, the only state reaching this limit after filtering, demonstrates adequate content coverage.

## Data preprocessing

Both datasets were preprocessed using the same process, which included the removal of redundant whitespace, non-essential punctuation, and peripheral content like headers and footers, while preserving sentence-level punctuation. The webpage content was segmented into samples comprising paragraphs paired with their preceding HTML headers to retain topical context. This produced cohesive information units suitable for annotation and IE.

## Annotation schema

We developed an annotation schema consisting of 26 binary (yes/no) labels, each formulated as a natural language question to support a question-answering (QA) framework. The initial label set was informed by qualitative content literature and refined through topic consolidation (e.g., combining throat cancer and genital cancer into "HPV causes cancer"). Labels were further refined through consultation with a health education researcher specializing in public health and cancer and a certified nurse midwife with expertise in women's health, reproductive care, and health education.

This QA-based formulation simplified annotation by reducing complex judgments to binary decisions and aligned with the QA instruction tuning of LLMs. Eleven labels were excluded from further experimentation as they were infrequent (<5 occurrences) in test and validation sets. Table 3 lists the remaining 15 labels with the associated yes/no questions.

Each label was accompanied by a detailed guideline, including clarifying descriptions and both positive and negative examples to reduce ambiguity and support consistent interpretation. These guidelines were iteratively refined during early annotation rounds. Full label descriptions are in Table 1 in S1 Table.

The Targeted HPV Dataset was annotated using Doccano by two independent annotators who marked relevant spans for each question [63]. The process was carried out in iterative batches, initially small to support training, and later larger for efficiency. Disagreements were resolved through discussion after each round, creating gold-standard labels. The final

**Table 3.** Annotation guideline summary, including the natural language question representing the binary labels and the label name.

| Question | Label |
| --- | --- |
| Does the text indicate HPV infection is asymptomatic? | Asymptomatic |
| Does the text indicate HPV infection cannot be cured? | No Cure |
| Does the text indicate HPV is primarily contracted through sexual contact? | Sexual Spread |
| Does the text describe HPV causing cervical cancer? | Cause of Cervical Cancer |
| Does the text indicate HPV vaccination can prevent cancers stemming from HPV? | Cancer Prevention |
| Does the text indicate HPV causing any cancer in general? | Cause of Cancer |
| Does the text indicate HPV vaccination can prevent spreading HPV? | Prevent Spread |
| Does the text indicate HPV vaccination as highly effective at preventing HPV and/or related cancers? | Highly Effective |
| Does the text describe the number of HPV vaccine doses that are needed? | Number Doses |
| Does the text describe safety concerns and potential side effects of HPV vaccination? | Side Effects |
| Does the text recommend HPV vaccination for children? | Rec-Children |
| Does the text recommend HPV vaccination for both males and females? | Rec-Male Female |
| Does the text recommend HPV vaccination? | Stance-Recommend |
| Does the text describe HPV vaccination as optional? | Stance-Optional |
| Does the text reference reputable sources (e.g., CDC, ACS, AAP)? | Source Credibility |

dataset achieved 0.87 F1 and 0.85 Cohen's Kappa Inter-Annotator Agreement. This dataset served as the benchmark for evaluating model performance.

## Prompting strategies

We explored three prompting strategies for text IE: 1) Zero-shot + Guidelines, 2) Zero-shot + Guidelines + Rationale, and 3) 1-shot + Guidelines + CoT prompting. Table 4 presents details of each strategy, incorporating short text samples. Each sample required 15 separate LLM queries based on our labels of interest, as combining questions reduced performance due to model confusion with longer tasks. All LLM implementations used 16-bit floating precision models. We use the term zero-shot to describe our setting because the task of predicting a text sample into our curated labels was never used to train the LLMs, even though the models may have been exposed to the data during pretraining for next-token prediction.

**Table 4. Prompting strategies.**

| Strategy | Prompt |
|---|---|
| Zero-shot + Guidelines | **System**: You are a helpful assistant<br>**Question**: Does the text describe HPV causing cervical cancer?<br>**Text**: HPV, a common virus, can lead to various health issues, including certain cancers like cervical, throat, and anal cancer.<br>**Guideline**: Relevant statements should explicitly indicate HPV can cause cervical cancer. References to other cancers are not applicable.<br>**Task**: Based on the provided text and guidelines, determine if the text answers the question. Please provide your answer with only a 'yes' or a 'no.' |
| Zero-shot + Guidelines + Rationale | **System**: You are a helpful assistant<br>**Question**: Does the text indicate HPV causing any cancer in general?<br>**Text**: HPV, a common virus, can lead to various health issues, including certain cancers like cervical, throat, and anal cancer.<br>**Guideline**: Relevant statements should explicitly indicate HPV can cause cervical cancer. References to other cancers are not applicable.<br>**Task**: Based on the provided text and guidelines, determine if the text answers the question. Provide a rationale (reasoning for your answer) and then the actual answer with only a 'yes' or a 'no.' |
| 1-shot + Guidelines + CoT | **System**: You are a helpful assistant<br>**Task**: You are given a text, and your task is to determine whether the text indicates HPV causing cervical cancer. Relevant statements should explicitly indicate HPV can cause cervical cancer. References to other cancers are not applicable.<br>Follow the example reasoning process provided below to help you answer this question. Follow the example reasoning process provided below to help you answer this question:<br>**Example:**<br>**example specific input text:** HPV, also known as Human Papillomavirus, has been identified as a primary cause of cervical cancer in women.<br>**example specific question:** Does the text describe HPV causing cervical cancer?<br>**Output:**<br>**CoT1:** *Does the text mention Cervical Cancer?*<br>**CoTAnswer1**: *yes*<br>**CoT2:** *Does the text assert a causal relationship between HPV and Cervical Cancer?*<br>**CoTAnswer2:** *yes*<br>**Final Answer:** *yes*<br>**Now, here's your input for the same task:**<br>**Text**: HPV, a common virus, can lead to various health issues, including certain cancers like cervical, throat, and anal cancer. |

**Zero-shot + Guidelines**: The LLM was prompted with a text sample, question, and paraphrased guidelines. These guidelines were adapted from the annotation protocol and iteratively refined based on relation performance with Llama 3.1 70B, using observed errors to improve clarity and effectiveness.\

**Zero-shot + Guidelines + Rationale:** This strategy extended the previous approach by first requiring the model to generate a rationale explaining its analysis before outputting a binary (yes/no) answer. This rationale was intended to expose intermediate reasoning steps to support downstream KD by transferring not just decisions but explanatory logic.

**1-shot + CoT:** This strategy used supervised CoT prompting, providing the model with a full exemplar consisting of a sample question-answer pair, a set of manually curated CoT sub-questions with corresponding answers, and the final decision [64]. Unlike model-generated CoT, these reasoning steps were human-authored and designed to reflect label-specific inference patterns. The structured reasoning was included in the KD process to convey granular step-wise reasoning. CoT subquestions were iteratively refined based on validation performance. Analysis showed that some labels (e.g., Asymptomatic) could be resolved with a single reasoning step, while others (e.g., Cause of Cervical Cancer) required multiple reasoning steps within a single CoT to arrive at the correct answer. CoT sub-questions were selectively incorporated into 7 of 15 labels where step-by-step reasoning showed clear gains. The full set of CoT sub-questions is presented in Table 2 in S1 Table.

## Knowledge distillation

We conducted KD experiments on the validation set using Llama models, selected for their QA performance and open-weight availability [65]. Llama 3.1 70B achieved performance approaching IAA under the Zero-shot + Guidelines strategy. However, its high computational cost made large-scale deployment impractical: labeling 1,000 samples took 36 hours on three A100 GPUs, with an estimated three weeks to process the full Comprehensive HPV Corpus. To address this challenge, we developed a KD framework to transfer the capabilities from the teacher model (Llama 3.1 70B) to smaller, more efficient student models (RoBERTa was selected upon experimental validation). We evaluated several student architectures (Llama 3.1 8B, 3.2 3B, 3.2 1B, RoBERTa and DistilBert), using the following distillation approaches:

**Encoder Baseline:** Applicable only to encoder models like RoBERTa and DistilBert, the approach frames the task as a multi-label classification problem. The model receives an input text sample and outputs a 15-dimensional vector corresponding to the binary labels. The student model was trained using the teacher model's predictions for the 15 binary labels.

**Encoder + CoT**: Extends Encoder Baseline by including CoT sub-questions as target labels. While input remains a text sample, output expands to a $(15 + m)$-dimensional vector, where $m = 7$ represents the total CoT sub-questions across all labels. Training follows the same procedure as the Encoder Baseline, with the model learning to predict both the original labels and the CoT sub-question responses.

**LLM Baseline:** For Llama students, this approach mirrors the Zero-shot + Guidelines strategy. The input includes a 7 text sample, a question, and paraphrased guidelines, and the model outputs a binary label. The student model was trained exclusively on the teacher model's binary predictions for each label.

**LLM + Rationale**: This approach extends LLM Baseline by requiring the model to generate a rationale before producing the binary label. The input consists of a text sample, a question, and paraphrased guidelines. The student model is trained on the teacher model's outputs, which include both the generated rationale and corresponding binary label. At inference, the input and output format mirrors the Zero-shot + Guidelines + Rationale strategy.

**LLM + CoT:** This strategy extends LLM + Rationale by replacing rationales with label-specific exemplars defining the supervised CoT reasoning. Each training instance includes a text sample, a question, paraphrased guidelines, and a

manually curated exemplar from 1-shot+CoT. The student model is trained to replicate the structured reasoning steps and binary label. At inference, inputs and outputs follow the 1-shot+CoT format.

Table 3 in S1 Table summarizes each student-teacher pairing, along with example inputs and outputs.

## Significance testing

To evaluate whether the performance differences between models were statistically significant, we applied the paired bootstrap test, generating 1000 pseudo test sets by sampling with replacement from the original dataset [66]. For each bootstrap sample (of size half the test set), we computed the difference in F1 scores between models and calculated a one-sided empirical p-value based on how often this difference exceeded the observed delta. This allowed us to identify the best-performing teacher and student models with statistical confidence at $p < 0.05$ (Confidence Interval of 95%), ensuring that observed improvements were unlikely to have occurred by chance.

Further details about our experimental paradigm are presented in S2 Text.

## Conclusion

This work addresses the critical challenge of systematically evaluating health information quality within a fragmented information ecosystem, using HPV information on DOH websites as a case study. NLP enables scalable analysis of such repositories, where manual evaluation would be resource-intensive. Instruction-tuned LLMs enable zero-shot and few-shot IE, reducing annotated data requirements, but their high computational costs hinder practical deployment. To address this challenge, we developed a resource-efficient pipeline combining a structured Q/A annotation schema, a curated dataset, and a KD framework that transfers capabilities from a large teacher model to a compact student model. We demonstrated this approach through a large-scale analysis of over 14,000 paragraphs from 48 DOH websites.

Future work could extend this research in several directions. First, the framework could support more detailed assessments, including cross-jurisdictional comparisons and analyses linking information availability to health practices and vaccination outcomes. Second, incorporating more samples into the KD process may improve student model performance. Third, extending the research to support multilingual content would broaden its applicability and help assess differences in topic coverage across language groups. Collectively, these enhancements would enable a more comprehensive understanding of how public health information is disseminated and where content may be improved.

## Supporting information

**S1 Text.**
(DOCX)

**S2 Text.**
(DOCX)

**S3 Text.**
(DOCX)

**S1 Table.**
(DOCX)

## Author contributions

**Conceptualization:** Saadat Hasan Khan.

**Data curation:** Saadat Hasan Khan.

**Formal analysis:** Saadat Hasan Khan.

**Investigation:** Saadat Hasan Khan.

**Supervision:** Kevin Lybarger.

**Validation:** Saadat Hasan Khan.

**Visualization:** Saadat Hasan Khan.

**Writing – original draft:** Saadat Hasan Khan.

**Writing – review & editing:** Saadat Hasan Khan.

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
