## [Decision Letter · Decision Letter 0]

9 Dec 2025

Response to Reviewers
Revised Manuscript with Track Changes
Manuscript
**Journal Requirements:**
**Additional Editor Comments (if provided):**
**Reviewers' Comments:**

**Comments to the Author**

1. Does this manuscript meet PLOS Digital Health’s publication criteria?

Reviewer #1: Yes

Reviewer #2: Yes

Reviewer #3: Yes

Reviewer #4: Yes

2. Has the statistical analysis been performed appropriately and rigorously?

Reviewer #1: Yes

Reviewer #2: Yes

Reviewer #3: Yes

Reviewer #4: Yes

3. Have the authors made all data underlying the findings in their manuscript fully available (please refer to the Data Availability Statement at the start of the manuscript PDF file)?

Reviewer #1: No

Reviewer #2: Yes

Reviewer #3: Yes

Reviewer #4: Yes

4. Is the manuscript presented in an intelligible fashion and written in standard English?

Reviewer #1: Yes

Reviewer #2: Yes

Reviewer #3: Yes

Reviewer #4: Yes

Reviewer #1: I am recommending Major Revision for this manuscript. This is a really interesting and timely study that tackles an important problem in public health. The knowledge distillation approach is clever, and I think the work has a lot of potential to be a strong contribution to the field. My main suggestions below are focused on strengthening the paper's alignment with open science principles and bolstering the interpretation of the case study.

This study presents a novel and efficient method for analyzing health communication content on state Department of Health (DOH) websites, using HPV information as a case study. The authors use large language models as a teacher to train a much smaller and faster student model (RoBERTa-Large) via knowledge distillation. The results show that the student model can perform the information extraction task nearly as well as the teacher model but is about 350 times faster. The authors then use their efficient model to analyze a large corpus of text from 48 DOH websites, presenting a heatmap of which HPV-related topics are covered in each state.

Major Comments

1. Data and Model Accessibility: The new annotated dataset is a fantastic resource for the community. To fully align with the open science goals of PLOS Digital Health and ensure other researchers can build on this important work, the authors should consider depositing the dataset and the final trained RoBERTa model in a public repository. The journal's policy requires data to be fully available before publication (please see the Data Availability policy).

2. Exploring the Data Collection Limits: The case study analyzing content across state DOH websites is a compelling application of the model. To make the conclusions from the heatmap even more robust, it might be helpful to expand the discussion on the potential impact of the Google Search API's 100-result limit. It's possible that for states with a lot of webpages, this limit might result in an underestimation of their topic coverage. Perhaps a small validation, like a deeper manual search for one or two states that appear to have low coverage, could help quantify this and really strengthen this part of the paper.

3. Expanding on the Error Analysis: It was interesting that the smaller RoBERTa model actually did better on some of the tasks that required more inference, while the larger teacher model was better at pulling out explicitly stated facts. I think the paper would be even more impactful if the authors could expand the discussion with their thoughts on why this might have happened.

Minor Comments

1. Typo (P11 L4): “Furthermore, out …”, out should be our.

2. For greater precision, the authors should consistently use a term like “topic coverage” instead of “information completeness” to accurately reflect that the analysis measures the presence of topics, not their quality or depth.

Reviewer #2: To ensure the claim that the model performs zero-shot classification, the authors should specify the time window of data collection for the HPV data provided by the Department of Health (DOH). The temporal gap between the model’s release date and the dataset collection period is critical to demonstrate that the model had no prior exposure to the collected data during pretraining. If the data were gathered before or near the model’s training period, there is no guarantee that portions of the data provided by DOH were not included in the model’s original training corpus. Therefore, the dataset collection must have occurred well after the release of the Llama 3.1 70B model to validate its characterization as a truly zero-shot evaluation.

I think this paper would have been better if the zero-shot concept had been excluded.

Reviewer #3: Th evaluated manuscript analyses a relevant and timely thematic at the link of public health communication and NLP. The research team proposed a well-organized knowledge-distillation framework that weights computational efficieicny and model performance, applying it convincingly to state Departament of Health platforms related to HPV. The utilized methodology sounds clear. I commend the team for collating reproducible annotation protocols, model comparison, as well as public-health related important features within a clear structure.

I believe the following areas must have a second look to improve the manuscript per si:

a) The discussion should better correlated computational findings with practical implications for health-related communication strategies as well as for vaccine uptake.

b) With regards to the ethical section, it would be appropriate to emphasize risk of bias propagation assessments in downstream systems.

c) I would call the authors attention to the data availability statement, particularly specificying the exact format and process for the dataset access to increase transparency and reproducibility.

d) Some sentences in the results are too lengthy and could be shortened for better clarity.

e) Clinically speaking, authors might emphasize the translational rhetoric through the articulation of how improved quality of health communication has been proven to influence measurable epidemiological parameters (including vaccination coverage)

f) As DOH platforms are utilized by contrasting groups, a short paragraph approaching disparities in digital access, health literacy, and cultural tailoring of online interactions would enrich the discussion.

g) This is critical - the authors should stress that the proposed framework could be definitely extended to NCD to assess the accuracy and impact of online health communication.

Reviewer #4: Efficient Information Extraction Using LLMs and Knowledge Distillation: A Study on HPV Health Communication

This manuscript presents a well-structured and technically sound study that leverages large language models (LLMs) and knowledge distillation (KD) to evaluate the quality and completeness of HPV-related information across U.S. state Department of Health (DOH) websites. The authors introduce a two-stage framework — first, developing a teacher–student model distillation approach using the Llama family and RoBERTa architectures, and second, applying the distilled model to large-scale web data. The study contributes to both computational methodology and public health informatics by providing an efficient, scalable approach for automated health communication assessment.

I have the following comments.

Major Comments

1. The paper provides an extensive overview of the data collection and annotation process, but a few methodological clarifications would improve reproducibility:

Dataset transparency: The Targeted HPV Dataset (n=400) is mentioned as available upon request, but a public link or structured data statement (e.g., schema of each variable, annotation format) would strengthen the data availability claim.

Annotation guidelines: While Appendix Table 1 provides questions and labels, inter-annotator agreement (IAA = 0.87 F1) lacks details on Cohen’s κ or whether disagreements were resolved via adjudication. Clarifying this would improve transparency.

Sampling bias: Since the dataset disproportionately represents primary pages, the authors should discuss potential sampling bias and how it might affect generalizability to broader DOH content.

2. The RoBERTa Large model (student) achieving near-parity with Llama 3.1 70B is impressive, yet the manuscript lacks comparison with other compact encoder-based models (e.g., DistilBERT, BioClinicalBERT, or PubMedBERT). A brief ablation or reference to prior work showing similar benchmarks would contextualize the advantage of the KD approach.

3. The use of bootstrap significance testing is commendable. However, please specify the confidence intervals (e.g., 95%) and provide effect sizes for key comparisons (Teacher vs Student). The F1 differences are small (Δ = 0.03); providing standard deviation or variance across labels would help readers assess robustness. It would be useful to visualize label-level performance variability using a bar or radar plot to show where the student model outperforms the teacher.

4. The qualitative error analysis section is insightful but could be expanded. Specifically:

Provide 2–3 concrete text examples illustrating why the student model performs better on inferred information (e.g., implicit causal relationships).

Include a short discussion of model interpretability (e.g., use of attention maps, SHAP, or LIME) to demonstrate how the model identifies key phrases or constructs during evaluation.

5. While the manuscript appropriately acknowledges ethical and legal constraints, two key areas merit deeper discussion:

Bias propagation: Beyond acknowledging potential model bias, a brief quantitative analysis (e.g., topic frequency or linguistic bias by state) would demonstrate proactive bias monitoring.

Data collection ethics: Some DOH websites restricted automated scraping (HTTP 403). The authors should state whether institutional review or data use compliance measures were taken, as per PLOS Digital Health’s transparency guidelines.

Minor Comments –

1.Introduction could be slightly condensed to improve flow. Several citations (e.g., refs. 11–17) could be grouped to streamline readability.

2. The use of “teacher” and “student” models should be defined clearly in the Methods section (first mention).

3. Some references (e.g., arXiv citations [43–47]) could be complemented by peer-reviewed publications where available.

4. Ensure consistency in F1 score reporting (some tables use two decimal places, others one).

**Do you want your identity to be public for this peer review?** For information about this choice, including consent withdrawal, please see our Privacy Policy

Reviewer #1: No

Reviewer #2: No

Reviewer #3: **Yes:** Israel Júnior Borges do Nascimento

Reviewer #4: No

**Figure resubmission:**

**Reproducibility:** To enhance the reproducibility of your results, we recommend that authors of applicable studies deposit laboratory protocols in protocols.io, where a protocol can be assigned its own identifier (DOI) such that it can be cited independently in the future. Additionally, PLOS ONE offers an option to publish peer-reviewed clinical study protocols. Read more information on sharing protocols at https://plos.org/protocols?utm_medium=editorial-email&utm_source=authorletters&utm_campaign=protocols

---

## [Editor Report · Decision Letter 1]

15 Feb 2026

Efficient Information Extraction Using LLMs and Knowledge Distillation: A Study on HPV Health Communication

PDIG-D-25-00609R1

Dear Mr. Khan,

We are pleased to inform you that your manuscript 'Efficient Information Extraction Using LLMs and Knowledge Distillation: A Study on HPV Health Communication' has been provisionally accepted for publication in PLOS Digital Health.

Best regards,

Dhiya Al-Jumeily OBE, PhD

Section Editor

PLOS Digital Health

**Additional Editor Comments (if provided):**

All reviewers' comments were addressed to a satisfactory level.